# Surgical Management of Upper Urinary Tract Urothelial Cell Carcinoma with Venous Tumor Thrombus: A Liver Transplant-Based Approach

**DOI:** 10.3390/jcm10245964

**Published:** 2021-12-19

**Authors:** Gaetano Ciancio, Marina M. Tabbara, Melanie Martucci, Jeffrey J. Gaynor, Mahmoud Morsi, Javier Gonzalez

**Affiliations:** 1Department of Surgery and Urology, Miami Transplant Institute, Jackson Memorial Hospital, University of Miami Miller School of Medicine, Miami, FL 33136, USA; 2Miami Transplant Institute, Jackson Memorial Hospital, University of Miami Miller School of Medicine, Miami, FL 33136, USA; mmt68@med.miami.edu (M.M.T.); mxm193355@miami.edu (M.M.); jgaynor@med.miami.edu (J.J.G.); mmorsi@med.miami.edu (M.M.); 3Department of Urology, Hospital General Universitario Gregoria Marañón, 28007 Madrid, Spain; fjgg1975@yahoo.com

**Keywords:** renal urothelial carcinoma, venous thrombus, radical nephroureterectomy and tumor thrombectomy, transplant-based techniques

## Abstract

Upper urinary tract urothelial cell carcinoma (UTUC) with venous tumor thrombus (TT) that extends into the renal vein (RV) and inferior vena cava (IVC) is a rare entity and its management is a surgical challenge. We report the largest single experience of surgical management of UTUC and accompanying venous TT with radical nephroureterectomy and tumor thrombectomy (RNATT) using transplant-based (TB) surgical techniques. From September 2003 to June 2021, nine patients with UTUC and venous TT underwent RNATT. Demographics, disease characteristics, surgical details, 30-day postoperative complications, and overall survival (OS) were analyzed. All nine patients had extension of the TT into the RV. Of those, seven had additional extension of the TT into the IVC. Venous TT level was categorized as 0 (*n* = 2), I (*n* = 2), II (*n* = 4), and IIIa (*n* = 1). Median tumor size was 12 cm (range 3–20 cm). Median estimated blood loss was 300 (range 150–1000) cc. One patient was still alive at last follow-up (4 months), and in total, eight patients have died with a median time-to-death of 12 months (range 10 days–24 months). RNATT using TB maneuvers like liver mobilization and pancreas-spleen en bloc mobilization provide excellent exposure to the retroperitoneal space and enable the safe removal of UTUC with venous TT.

## 1. Introduction

Renal cell carcinoma (RCC) is the most common malignancy that results in venous tumor thrombus (TT) and is associated with vascular extension into the inferior vena cava (IVC) in 4–10% of cases [1]. Upper urinary tract urothelial cell carcinoma (UTUC) is a rare urological malignancy with aggressive behavior, and its extension into the renal vein (RV) and IVC is an even rarer occurrence, with a total of only 49 such patients previously being reported in the literature [2]. In a large study of 2380 patients (from 101 centers in 29 countries) with UTUC, 1430 (60.1%) underwent radical nephroureterectomy, and none had TT venous extension [3]. In another series of 102 patients with transitional cell carcinoma (TCC), 5 patients (5%) had venous TT, and one patient had level IV TT [4]. Thus, while the optimal surgical management of these complex tumors has not been well documented, surgery remains as the only potential cure for these patients [5,6].

Herein, we describe the results for nine cases of UTUC and venous TT managed with radical nephroureterectomy and tumor thrombectomy (RNATT) using transplant-based (TB) surgical techniques. These methods aid in achieving safe resection of large urological tumors with venous TT, reducing blood loss and blood transfusion requirements by providing adequate exposure and vascular control of the IVC [1,7,8].

## 2. Materials and Methods

From September 2003 to June 2021, nine patients with UTUC and accompanying venous TT underwent RNATT at our institution. The study of these patients was in accordance with our University of Miami Institutional Review Board (ethics committee approval code MOD00044523) and the Helsinki Declaration (as revised in 2013). Informed consent was given, including the risks associated with the complexity of the surgery, such as bleeding and tumor embolization into the pulmonary arteries.

Initial diagnosis was made using computed tomography (CT) (Figure 1). Cardiac, renal, and respiratory status were evaluated preoperatively. The thrombus level was confirmed in five patients with magnetic resonance imaging (MRI). Cranial extent of the tumor was defined by Neves et al. [9] and our own classification for RCC with TT [10]. Clinical and pathological staging were performed using the TNM classification. Preoperative embolization was not performed in any of the patients.

Transesophageal echocardiography (TEE) was used in all patients, as it is considered to be an essential part of their surgical management. TEE is used in both delineating the cranial extent of the TT and in identifying/ruling out the existence of pulmonary emboli before, during, and after surgery [11]. 

Overall survival (OS) was ascertained by a review of the patient’s medical record or the Social Security Death Index (SSDI) database when necessary [12].

### 2.1. Operative Technique

This TB surgical technique for resection of RCC and UTUC with tumor thrombus has been described at length previously and is illustrated in Figure 2 [5,6,8,13]. First, a subcostal incision was made approximately two fingerbreadths below the right or left costal margin (according to the tumor location), extending out laterally to the mid axillary line. Self-retaining retractor was then placed, elevating the costal margins and splaying them laterally toward the axillae.

We pursued early intraoperative ligation of the involved renal artery, which decompresses collateral circulation, decreases blood loss, and prevents the need for blood transfusions [14]. With the transabdominal approach, exposure of a left UTUC began by mobilization of the descending colon [5,8]. The spleen was dissected off the diaphragm and mobilized en bloc with the pancreas toward the midline. This exposed the entire upper retroperitoneal space from the diaphragm to the superior border of the kidney [8]. 

In some cases, exposure of a right UTUC required liver mobilization [5,6,8,10,13]. This began with the dissection and division of the ligamentum teres. The falciform ligament was divided with cautery, and this incision was carried around the right superior coronary ligament, bypassing the left side, and dividing the left triangular ligament. The visceral peritoneum to the right of the hepatic hilum and the infrahepatic vena cava was incised in conjunction with the right inferior coronary and hepato-renal ligaments. The liver is then gradually rolled to the left [5]. Liver mobilization allows for adequate exposure of the upper abdomen and retroperitoneum and facilitates removal of a large UTUC [10]. For the level IIIa (retrohepatic) TT, an opening in the lesser omentum allows the porta hepatis to be controlled with a Rummel tourniquet; a Pringle maneuver can then be carried out (temporarily occluding the portal venous and arterial inflow to the liver) as required [10]. We then proceeded with the ‘piggyback’ liver transplant technique [5,15]. The term “piggyback” is used because the recipient’s vena cava is left in situ and the liver is mobilized off the vessel. Small hepatic veins passing from the right and caudate lobe are ligated and divided. The liver is dissected off the inferior vena cava until it lies in a piggyback fashion, attached to the inferior vena cava only by the major hepatic veins.

In this fashion, the infrahepatic and retrohepatic portions of the IVC are completely exposed (level IIIa TT). In addition to mobilizing the liver off the cava, creating a plane between the IVC and the posterior abdominal wall was also important because it permitted circumferential vascular control of the cava in case of the need to perform the reconstruction or the removal of the IVC. In addition, small tributaries can become engorged to look like lumbar vessels, and they need to be identified and ligated. 

The infrarenal cava was controlled and a vascular clamp was placed across the retrohepatic IVC below the major hepatic veins (level IIIa TT). The right adrenal and RV were clamped for the left UTUC. The IVC was incised, and the tumor was sharply dissected off the IVC. Following removal of the tumor thrombus, the IVC was sutured closed. The rest of the involved ureter was removed along with the kidney.

### 2.2. Chemotherapy

Chemotherapy was initiated within 90 days of surgery; none of the patients included in this series received neoadjuvant chemotherapy. Chemotherapy was administered according to the following schedule: four 21-day cycles of chemotherapy with either cisplatin (70 mg/m^2^) or carboplatin (if glomerular filtration rate < 50 mL/min) administered intravenously on day 1, and gemcitabine (1000 mg/m^2^) administered intravenously on days 1 and 8. 

### 2.3. Intraoperative and Postoperative Variables

The intraoperative variables, including operative time, blood loss, blood transfusion requirements, identification of pulmonary emboli, and death, were recorded. Postoperatively, cardio-pulmonary complications, deep vein thrombosis, extended ileus, length of hospital stay, renal function, and tumor histology were noted.

### 2.4. Statistical Methods

Medians and ranges of values for the selected variables were reported as descriptive statistics for the patients in this study. Actuarial survival following surgery was estimated using the Kaplan–Meier technique.

## 3. Results

Median age was 72 (range 59–82) years, and six of the patients were female. All nine patients had extension of TT into the RV. Of those, seven had additional extension of the TT into the IVC. None of the patients had TT extending into the right atrium. The IVC TT level was classified as 0 (*n* = 2), I (*n* = 2), II (*n* = 4), and IIIa (*n* = 1). Median tumor size was 12 cm (range 3–20 cm). Median estimated blood loss was 300 (range 150–1000) cc. Four patients required a blood transfusion. Median requirement of packed red blood cells was 0 (range 0–7) U during surgery. The median postoperative creatinine was 1.0 mg/dL (range 0.6–1.4 mg/dL). The median length of hospital stay was 8 (range 6–28) days. 

Two of these nine cases had a complex IVC reconstruction. One patient with a large renal mass and level II TT required a radical nephroureterectomy with reconstruction of the IVC using a ringed polytrafluoroethylene (Gortex) graft and the insertion of an IVC filter [13] (Figure 3A). Another patient with right UTUC and level IIIa TT required resection of the obstructed IVC and a segment of the left RV. The remaining portion of the IVC below the major hepatic veins was anastomosed to the remaining left RV, and the distal IVC was not reconstructed (Figure 3B). All of the patients had a high-grade urothelial carcinoma, and one patient had squamous differentiation observed in 40% of the tumor. None of the patients had muscle invasive urothelial carcinoma and no concomitant procedures were performed. Eight patients had lymph node metastasis. One patient had adrenal metastasis. Two patients had documented distant metastasis before surgery. One patient with documented lung metastasis before surgery died in the immediate postoperative period (at 10 days post-surgery) of respiratory failure and sepsis. Another patient had documented liver metastasis before surgery. None of the patients developed DVT, but two patients developed pulmonary emboli at 5 and 7 days after surgery, respectively. No other postoperative complications were observed. 

Three of these patients received adjuvant chemotherapy. Three other patients who were being followed by an oncologist did not receive chemotherapy due to advanced age (range: 72–82 years). In total, eight of the nine patients have died. Median survival was 12 months (range: 10 days–24 months); only one patient was still alive at last follow-up (4 months).

## 4. Discussion

Patients with invasive UTUC do not constitute a substantial portion of cancer cases and their treatment can be very complicated. A review of the MD Anderson Cancer Center’s experience found that there has been minimal improvement in disease-specific survival of UTUC, calling for a change in treatment regimen [16]. Although efforts have been made to implement multidisciplinary treatment of UTUC, including neoadjuvant and adjuvant chemotherapy treatments and perioperative immunotherapy [17], surgery is currently the only viable treatment option to possibly cure patients with UTUC and accompanying venous TT.

Although the surgical procedure for patients presenting with invasive or metastatic UTUC is high-risk and challenging, it has several advantages. One is that it provides better and more accurate staging than the combination URS biopsy/cross-sectional imaging, thus paving the way for further treatment decision-making [18]. It decreases tumor burden and provides for better immunological response in the treated patient, who cannot fight such an amount of neoplastic disease effectively from an immunological standpoint, particularly if immune checkpoint inhibitors are planned. It also provides relief and palliation in cases of flank pain, hematuria, or paraneoplastic syndromes, which is quite frequently reported in the subpopulation [19]. Last, but not least, it increases the overall survival and cancer-specific survival [20], particularly if combined with an adequate lymph node dissection [21].

This current study is the largest series of UTUC with TT resected safely and without the use of large amounts of blood products due to the early ligation of the renal artery. Two of these nine cases had a complex IVC reconstruction. One patient had an interposition ringed Gortex graft with an IVC filter [22], and the other patient had the proximal IVC anastomosed to the left RV but without reconstruction of the distal IVC. The IVC was completely obstructed in these two cases, with the occurrence of a distal blood thrombus being a distinct possibility [23].

Our experience affirms the safe use of liver transplantation techniques for the resection of UTUC tumors with TT extending into the IVC. Clearly, this approach facilitates the resection of large UTUC tumors by increasing the exposure of the retroperitoneal space. The concept of resorting to an entirely intra-abdominal approach without the need (in most cases) for cardiopulmonary or veno-venous bypass is an additional byproduct of this approach [1,5,8], as cardiopulmonary or veno-venous bypass is known to increase the patient’s risk of experiencing postoperative morbidity, particularly blood loss. However, in cases where tumor thrombus extends into the right atrium, it cannot always be possible to surgically manage patients by only a transperitoneal approach. Supradiaphragmatic thrombi are managed via a median sternotomy approach, providing exposure for the institution of CPB and access to the suprahepatic and retrohepatic IVC as necessary [24]. 

Neoadjuvant and adjuvant chemotherapy before or after radical nephroureterectomy (RNU) has been used to improve the prognosis of UTUC; however, the evidence remains mixed on their efficacy [17]. Neoadjuvant chemotherapy (NC) has been shown to result in lower disease recurrence and mortality rates compared to RNU alone without compromising the use of definitive surgical treatment [25]. There is a strong argument for the use of neoadjuvant, as opposed to adjuvant, chemotherapy, as it spares the renal function following surgical intervention [26]. On the other hand, the rationale for adjuvant chemotherapy (AC) is clear to most in the urologic oncology community. Patients are treated based on the most accurate pathologic staging obtained from RNU specimens, which prevents overtreatment. Additionally, there is no delay in patients receiving definitive surgical treatment, and they do not undertake the risk that their disease may not respond to chemotherapy [27].

There is no doubt that our cases are locally advanced metastatic cases according to cross-sectional preoperative imaging. In this way, we would not overtreat the patient if this would have been our therapeutic schedule choice, and we would obtain better margins and theoretically better oncological outcomes. On the contrary, the surgery should not be delayed. The time frame until surgery in our cases of metastatic UTUC is crucial to avoid the potential catastrophe of the thrombus extending upwards through the IVC and right atrium, making surgery more difficult or even impossible. The most common therapeutic schedule is cisplatin–gemcitabine (a total of four 21-day cycles) in patients with normal GFR or carboplatin (or other renal sparing schedules) if GFR is proved decreased. Patients unfit for chemotherapy (due to age, comorbidity, or a combination of factors) will not receive additional treatment after surgery, but they will still have the benefit for palliation and symptoms relief [28].

Despite aggressive surgical treatment, the prognosis of these patients is very grim, with reported survival being between 6 months and 14 months [2,29]. Huber et al. reported a median survival of 8.9 months for five patients having renal TCC and venous TT; all five of these patients died of their disease [4]. Thus, in spite of our being able to perform a radical nephroureterectomy with minimal complications observed in all patients, our observed median patient survival of 12 months (range: 10 days–24 months) did not appear to differ from these other reports of UTUC with TT [2,4,29,30]. 

The main limitation of this study could be represented by the potentially low reproducibility of this procedure among the entire urological community. As the surgical management of UTUC with tumor thrombus is very complex, treatment should be decided within a multidisciplinary consensus.

## 5. Conclusions

TB approaches that utilize liver, pancreas-spleen, and IVC mobilization techniques help to achieve a good exposure of the retroperitoneal space and excellent control of the IVC for the safe resection of UTUC with venous TT; however, disease-specific survival is still poor in these patients.

## Figures and Tables

**Figure 1 jcm-10-05964-f001:**
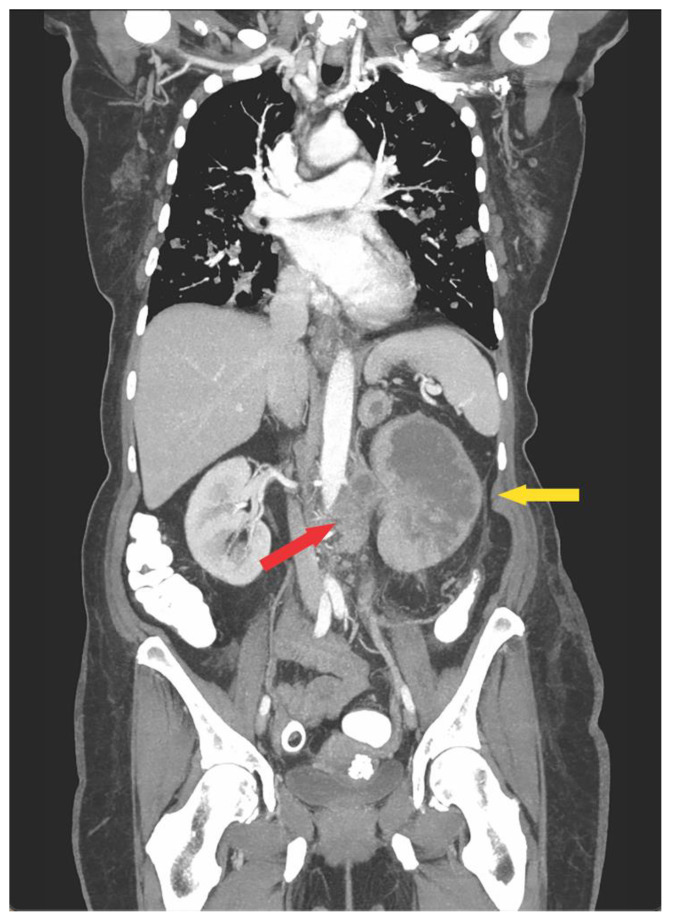
Computed tomography scan showing a renal mass infiltrating the left kidney (yellow arrow), also demonstrated a lymph node over the aorta (red arrow).

**Figure 2 jcm-10-05964-f002:**
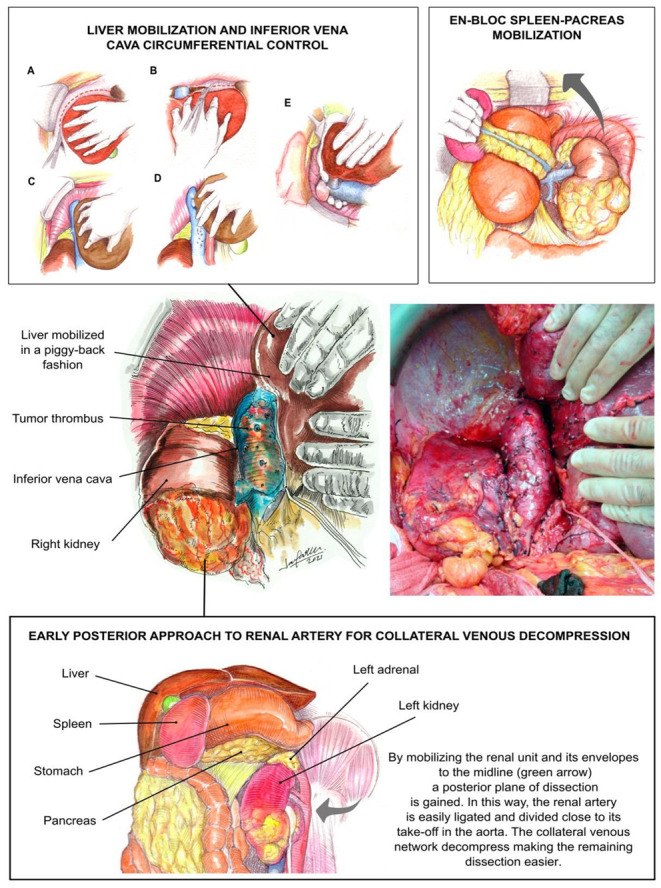
Transplant-based approach to radical nephroureterectomy in conjunction with tumor thrombectomy. Central figures show the complete exposure of the infrahepatic and retrohepatic segments of the inferior vena cava in the patient harboring a level IIIa tumor thrombus. The liver is mobilized commencing by dividing the left and right triangular ligaments (**A** and **B**, respectively). Once the right coronary ligament is divided, the right hepatic lobe can be gradually rolled to the midline (**C**). Piggy-back liver dissection requires the control and division of all the short veins communicating the right and caudate lobes with the anterior aspect of the inferior vena cava (**D**). In addition, the inferior vena cava is controlled circumferentially by detaching its posterior aspect from the posterior body wall (**E**). Further dissection is facilitated by early ligation of the main renal artery of the kidney involved. Early ligation is accomplished by gaining a posterior plane of dissection and mobilizing the entire kidney to the midline. Ligation is performed close to the take-off of the renal artery in the aorta. By ligating the renal artery, the collateral venous network generated in response to caval occlusion decompress, thus making the dissection less prone to bleed.

**Figure 3 jcm-10-05964-f003:**
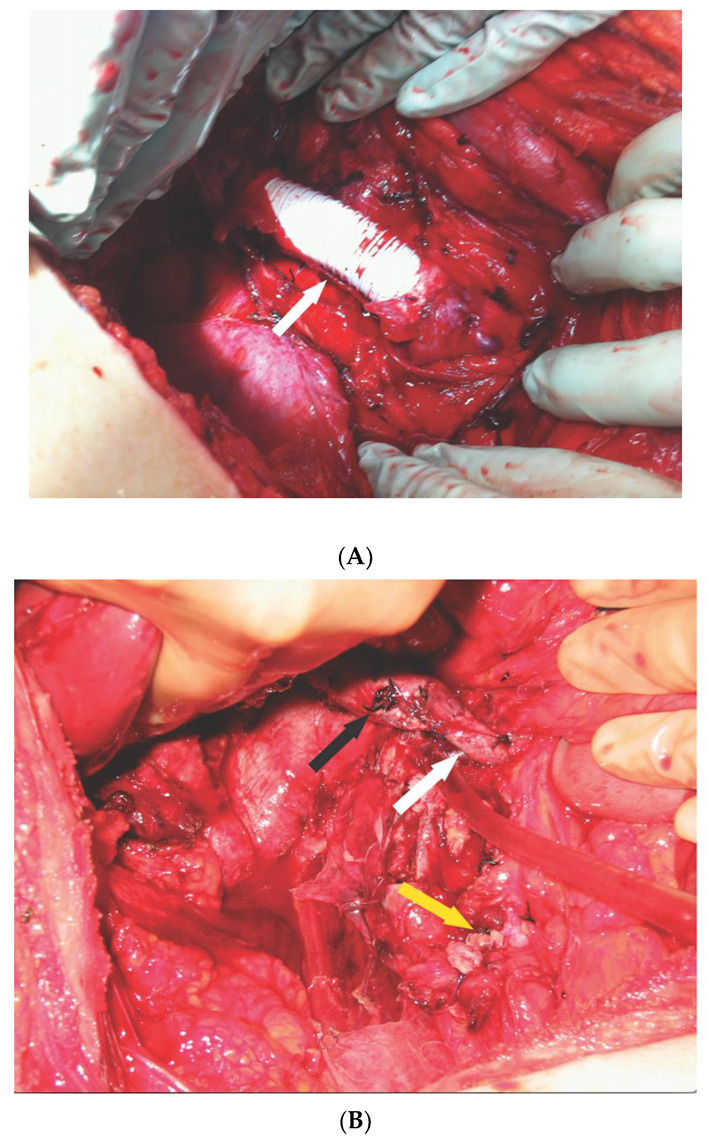
(**A**), Inferior vena cava (IVC) reconstructed with ringed polytrafluoroethylene graft (white arrow) with IVC filter placed inside the graft; (**B**) Anastomosis of the proximal IVC (black arrow) anastomosed to remnant of the left renal vein (white arrow), distal IVC was oversewn (yellow arrow).

## Data Availability

The data presented in this study are available on request from the corresponding author. The data are not publicly available due to privacy issues.

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
