# Peer review of "Surgical Management of Upper Urinary Tract Urothelial Cell Carcinoma with Venous Tumor Thrombus: A Liver Transplant-Based Approach"

_jcm, 2021, doi:10.3390/jcm10245964_

Round 1

Reviewer 1 Report

In this case series, Ciancio et al. described the outcomes of nine patients with Renal Urothelial Carcinoma (RUC) associated with tumor thrombosis managed with radical nephroureterectomy and tumor thrombectomy (RNATT) using transplant-based surgical techniques. The authors pointed out as these methods aid in achieving safe resection of tumor associated thrombosis minimizing intraoperative complications and/or adverse events by increasing the exposure of the retroperitoneum space. 

Overall interesting technique for a challenging cytoreductive surgery as presented with these cases with a rare and aggressive tumor. Moreover, as the authors stated, surgery is currently the only viable option in such a setting.

Please pay attention to the following recommendations for revision.

Although very detailed, main limitation of this study could be represented by the potentially low reproducibility of this procedure among the entire urological community. On the other hand, this case series indirectly pointed out as the management of this disease should be decided within a multidisciplinary consensus in a referral Centre. The authors could add this to the Discussion.

Renal Urothelial Carcinoma (RUC) should be defined as Upper Urinary Tract Urothelial Cell Carcinoma (UTUC) involving the renal pelvis and/or renal calyces. Please refer to UTUC in the whole manuscript as this definition is more generalizable.

Any information about concomitant procedures such as cystectomy and urinary diversion? Since these entities are sometimes associated with synchronous muscle-invasive bladder cancer, presence of concomitant procedures should be clarified.

Author Response

Point 1: Although very detailed, main limitation of this study could be represented by the potentially low reproducibility of this procedure among the entire urological community. On the other hand, this case series indirectly pointed out as the management of this disease should be decided within a multidisciplinary consensus in a referral Centre. The authors could add this to the Discussion.

Response 1: Thank you for this suggestion. We agree with this limitation and have included this statement in the discussion (lines 368-371)

Point 2: Renal Urothelial Carcinoma (RUC) should be defined as Upper Urinary Tract Urothelial Cell Carcinoma (UTUC) involving the renal pelvis and/or renal calyces. Please refer to UTUC in the whole manuscript as this definition is more generalizable.

Response 2: We agree with this suggestion and have now referred to Renal Urothelial Carcinoma (RUC) as Urothelial Cell Carcinoma (UTUC) throughout the manuscript. 

Point 3: Any information about concomitant procedures such as cystectomy and urinary diversion? Since these entities are sometimes associated with synchronous muscle-invasive bladder cancer, presence of concomitant procedures should be clarified

Response 3: Muscle-invasive bladder cancer was not present in the cases we describe and no additional procedures were performed during the RNATT. This is clarified in the Results (lines 161-163).

Reviewer 2 Report

Dear Editor,

In this retrospectively analysed case series, the authors reported their surgical technique adapted from liver transplantation for the management of renal urothelial cancer with tumor thrombus extending into inferior vena cava which is a rare but severe disaese.

The outcomes of the surgical technique were satisfactory even if the oncological consequences are worse due to the nature of this aggressive disease, as expected.

Below, I have some criticisms for the manuscipt.

  • At first sight, the title of the manuscript could not help the reader to distinguish which transplantation procedure it addresses. (liver, kidney etc.?)

For example, urologists who are the main target readers of this manuscript may understand 'kidney transplant based surgical management' at first reading which is not the message of the manuscript.

I recommend the authors to revise the title of the manuscript as defining hepatic transplantation procedure clearly.

  • I recommend the authors to draw a 2 figures highlighting the superiority of piggyback technique than the other technique. (Figure 1) the piggyback technique and 2) the commonly used other technique)

  • Did any of the cases have thrombus extending into right cardiac atrium?

If no, the authors should state as 'none of the patients had thrombus extending intoto right atrium'.

Because it can not always possible to manage by only transperitoneal approach when thrombus extends into the right atrium. Sternotomy and mediastinal approach needs in such situations. The readers have to be aware about this entity. The authors may write 2-3 sentences in discussion section to increase the awareness of these approaches.

Author Response

Point 1: At first sight, the title of the manuscript could not help the reader to distinguish which transplantation procedure it addresses. (liver, kidney etc.?)

For example, urologists who are the main target readers of this manuscript may understand 'kidney transplant based surgical management' at first reading which is not the message of the manuscript.

I recommend the authors to revise the title of the manuscript as defining hepatic transplantation procedure clearly.

Response 1: We agree with this suggestion to specify the liver transplant techniques used in the surgical management of UTUC with TT.  We have revised the title of the manuscript to Surgical Management of Upper Urinary Tract Urothelial Cell Carcinoma with Venous Tumor Thrombus: A Liver Transplant-Based Approach.

Point 2: I recommend the authors to draw 2 figures highlighting the superiority of piggyback technique than the other technique. (Figure 1) the piggyback technique and 2) the commonly used other technique)

Response 2: Thank you for this suggestion. We have included a figure containing the main advantages of our technique regarding exposure at the upper abdominal quarters (particularly at diaphragmatic domes with liver mobilization and spleen-pancreas mobilization),  and vascular control (posterior access to the renal artery to decrease collateral circulation before attempting to develop the anterior plane of dissection, and circumferential control of the IVC).

We also included a figure of the piggyback technique, as well as couple of diagrams containing the reconstruction of the IVC that carried out in n=2 cases.

We believe that other techniques are a call of other authors, therefore we will not be including figures regarding other techniques.

Point 3: Did any of the cases have thrombus extending into right cardiac atrium? If no, the authors should state as 'none of the patients had thrombus extending into right atrium'.

Response 3: None of the cases involved tumor thrombus extension into the right atrium. This clarification can be found in the Results (line 148)

Point 4: Because it cannot always be possible to manage by only transperitoneal approach when thrombus extends into the right atrium. Sternotomy and mediastinal approach needs in such situations. The readers have to be aware about this entity. The authors may write 2-3 sentences in discussion section to increase the awareness of these approaches.

Response 4: Thank you for this suggestion. We agree that a median sternotomy approach should be discussed as the surgical management for UTUC with supradiaphragmatic thrombi. We discussed this entity in the Discussion (lines 259-263).

Reviewer 3 Report

In this article, authors described the outcomes of 9 patients with upper tract urothelial carcinoma associated with tumor thrombosis managed with radical nephroureterectomy and tumor thrombectomy using a transplant-based surgical techniques. The authors already described this technique in different previous researches. 

In terms of specific oncological outcomes it is not clear what can be the value of this type of surgery. It is not clear the type and timing of chemotherapy. In general, the surgical technique is really interesting but I believe that surgery needs to be extensively discussed for this type of tumors. Please add more info about the chemotherapy and add a paragraph in the discussion to support the use of surgery for upper tract urothelial carcinoma.

Author Response

Point 1: In terms of specific oncological outcomes it is not clear what can be the value of this type of surgery. It is not clear the type and timing of chemotherapy. In general, the surgical technique is really interesting but I believe that surgery needs to be extensively discussed for this type of tumors.

Response 1: Thank you for your suggestions. The treatment of invasive and metastatic UTUC is tough due to the lack of effective protocols and guidelines, and the recommendations for chemotherapy are still controversial. We believe it is important to clarify this in our manuscript, therefore we further discussed the type and timing of chemotherapy, as well as the surgical interventions for these types of tumors. These changes are addressed below.

Point 2: Please add more info about the chemotherapy.

Response 2: Only 3 patients received chemotherapy after surgery and the regimen is described in a newly added section of the Methods (line 129). We also discussed the use of neoadjuvant or adjuvant chemotherapy in UTUC treatment (lines 264-275), as well as the benefit of adjuvant chemotherapy for our cases (lines 276-360).

Point 3: Please add a paragraph in the discussion to support the use of surgery for upper tract urothelial carcinoma.

Response 3: Surgery in the setting of invasive and metastaic UTUC has several advantages, of which we have further elaborated on in the Discussion (lines 227-245)

Round 2

Reviewer 1 Report

No further comments. Congratulations for this nice work.

This manuscript is a resubmission of an earlier submission. The following is a list of the peer review reports and author responses from that submission.